# Control-Based 4D Printing: Adaptive 4D-Printed Systems

**Ali Zolfagharian** [1,*] , **Akif Kaynak** [1] , **Mahdi Bodaghi** [2] , **Abbas Z. Kouzani** [1] , **Saleh Gharaie** [1] **and Saeid Nahavandi** [3]

[1] School of Engineering, Deakin University, Geelong 3216, Australia; akaynak@deakin.edu.au (A.K.); abbas.kouzani@deakin.edu.au (A.Z.K.); s.gharaie@deakin.edu.au (S.G.)

[2] Department of Engineering, School of Science and Technology, Nottingham Trent University, Nottingham NG11 8NS, UK; mahdi.bodaghi@ntu.ac.uk

[3] Institute for Intelligent Systems Research and Innovation (IISRI), Deakin University, Geelong 3216, Australia; saeid.nahavandi@deakin.edu.au

\* Correspondence: a.zolfagharian@deakin.edu.au

**Abstract:** Building on the recent progress of four-dimensional (4D) printing to produce dynamic structures, this study aimed to bring this technology to the next level by introducing control-based 4D printing to develop adaptive 4D-printed systems with highly versatile multi-disciplinary applications, including medicine, in the form of assisted soft robots, smart textiles as wearable electronics and other industries such as agriculture and microfluidics. This study introduced and analysed adaptive 4D-printed systems with an advanced manufacturing approach for developing stimuli-responsive constructs that organically adapted to environmental dynamic situations and uncertainties as nature does. The adaptive 4D-printed systems incorporated synergic integration of three-dimensional (3D)-printed sensors into 4D-printing and control units, which could be assembled and programmed to transform their shapes based on the assigned tasks and environmental stimuli. This paper demonstrates the adaptivity of these systems via a combination of proprioceptive sensory feedback, modeling and controllers, as well as the challenges and future opportunities they present.

**Keywords:** control-based; 4D-printing; adaptive; 4D-printed systems

## 1. Introduction

The market share of additive manufacturing (AM) in the global industry continues to grow, reaching $7 billion USD in 2017 and estimated to hit $33 billion USD by 2023 [1]. Three-dimensional (3D) printing technology has introduced new pathways to create objects with complex geometries one layer at a time. This method has several advantages over traditional manufacturing processes such as faster production, unrestricted design, ability to incorporate multiple materials into a single item in a single step fabrication process with a reduced cost and waste [2]. However, 3D printing is yet to evolve to produce intricate dynamic structures with controllable dimensions. Assembling stimuli-responsive materials, including polymers, hydrogels, alloys, ceramics and composites, and incorporating their functions inside a single printed construct has recently emerged under the label of four-dimensional (4D) printing. The fourth dimension refers to the dynamic response of the 3D-printed structure in response to a controlled stimuli such as ohmic parameters (current and voltage), water (absorption), heat (pre-strain), electromagnetic radiations (infrared), magnetic field and pH. The 4D printing can biomimic natural processes in self-constructing structures in automotive and aerospace industries, drug delivery and medical devices, soft robotics and other engineering applications [3–10].

Almost all of the current 4D-printed constructs solely respond to external stimuli with little adaptability to dynamic changes and uncertainties [11]. However, control-based 4D printing enables adaptive time-dependent spatio-temporal regulation in response to external stimuli while the 4D-printed systems interacts with uncertain environments [12–14]. Schematic diagrams of control-based 4D printing in adaptive 4D-printed systems are presented and discussed here with diverse applications as shown in Figure 1. These systems were developed by adding 3D-printed sensors into 4D printing, monitored by a controller. In other words, the adaptive 4D-printed systems consist of three components, namely, 3D-printed funcational layers, including embedded sensors and actuators and a controller. Indeed, there are modules required as peripheral and interfaces such as stretchable electronic networks boards [15–17] and 3D-printed batteries [17–20] that have already been broadly investigated and reported [6,17,21].

Controllable shape-changing 4D-printed systems have potential applications in tissue engineering to improve the functionality of the generated tissue [22–24]. Currently, an external mechanical stimuli, e.g., strain, is applied to the cells' seeded scaffold in a bioreactor. However, this could be achieved by 4D printing the scaffold, made from shape-memory polymers with intrinisically mechanical stimulus [22–24]. In chemotrapy, there is a high risk of inefficient treatment and a high risk of side effects when the drug is released in unintended organs and sites of the body [25]. Adaptive 4D-printed systems could be utilized in such scenarios to control the drug release to reach the intended site based on specific environmental circumstances [24]. Bandgapping a certain frequency region [26] and absorbing the energy of metastructures are also prospective applications of adaptive 4D-printed structures via control-based 4D printing [27]. Soft robotics is another interesting application where the adaptability of 4D printing can be utilized to print and control constructs for grasping, sorting and handling frail objects. Such systems could be used in regulating the fluids in microfluidics, as 4D-printed valves and pumps [28].

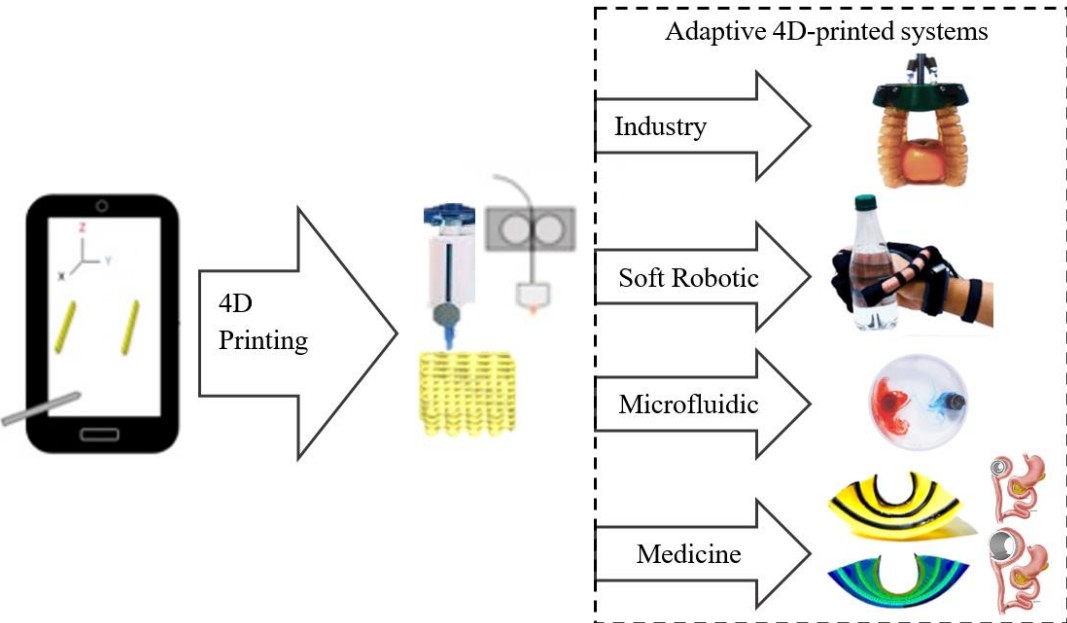

**Figure 1.** Some applications of the adaptive four-dimensional (4D)-printed systems. The images from top right to bottom are reproduced from [29] with the permission of the MDPI, [30] with the permission of the RMIT University Research Repository; Reproduced with permission from [31], Copyright Elsevier, 2015; [32] with the permission of the MDPI.

This paper consists of four sections as follows: the introduction of adaptive 4D-printed systems and the various methods to incorporate control strategies into 4D printing is described in Section 2. It is followed by analysing the integration of 3D-printed sensors into 4D printing in Section 3. This section explores different types of sensors that could be used in control-based 4D printing to provide the proprioceptive and environmental feedback information. In Section 4, the adaptive 4D-printed systems design is explained. Section 5 concludes the paper by presenting the challenges and future opportunities of these systems.

## 2. Controllable 4D-Printed Systems

One of the early representations of controllable 4D-printed systems was brought about by embedding off-the-shelf flexible sensors into 3D-printed fluidic actuators [7]. Subsequently, conductive hydrogel inks were developed as tactile and force sensors to measure the external load on a 4D-printed pneumatic actuator [33]. The flexible sensors were designed based on the resistive principle and were utilized to measure and control the bending angle of a 4D-printed pneumatic actuator [34]. Ionic polymer–metal composites (IPMCs) have also been utilized as both sensors and actuators in 4D-printed systems [35–37]. A controllable 4D-printed paper was recently fabricated by a composite polylactic acid (PLA)–graphene filament. The system used a resistive heating element as an actuator and electric potential as a sensor [38].

There have been some advances in the application of controllable 4D-printed systems in the biomedical sector. A 4D-printed system was developed to measure real-time blood pressure and the release of a drug in a controlled manner [39]. A 4D-printed artificial skin was made of thermo-responsive ionic hydrogel which used the capacitive principle for sensing and fiber enlargement as actuation [40]. A combination of poly (D-lactide) (PDLA) and polycaprolactone (PCL) was used to 4D print a biodegradable and biocompatible smart stent [41]. The PCL was used to fabricate an airway stent using the melt streolithograpy (SLA) technique. The stent shape was designed according to a patient's tracheobronchial track and delivered in its temporary state. An educated approach for the PCL molecule weight selection was used to control the shape-memory behavior of PCL including the strain fixity rate and the strain recovery rate. Once it was implanted the stent was thermally actuated and recovered to its original shape (set shape memory) providing structural support and preventing the trachea from collapsing.

In tissue engineering, the mechanical stimulus of cell-seeded scaffolds plays a key role in cell proliferation and differation processes [42], in which a biroreactor is required to apply the mechanical forces. However, by the 4D printing of shape-memory polymers (SMPs) based scaffold, the mechanical forces are applied by the time-controlled deformation of SMPs due to an external stimulus such as thermal radiation. The effectiveness of this concept was investigated using shape-memory polyurethane to fabricate 3D scaffolds [22]. It was shown that such a technique has a clear effect on cell alignment and could be a robust alternative to bioreactors in tissue engineering.

The controllable 4D-printed system found a novel application in soft robotics manufacturing through an embedded 3D-printed sensor [43]: a liquid conductor printed as tactile sensor channels to measure the amplitude and location of external forces. Machine-learning (ML)-based control algorithms were developed and applied to enhance the autonomy of 4D-printed soft robots in the assigned tasks [44]. A fusion of different sensors was co-printed in a single step to enable 4D-printed systems with the adaptation to external stimuli. To achieve this, a composite of thermoplastic polyurethane and carbon black was printed by the fused deposition modeling (FDM) as pressure and position sensors operated based on a piezo-resistive principle in a 4D-printed pneumatic actuator [44].

Not all the 4D-printed systems developed thus far have relied on sensory feedback information to function. There have been recent developments of 4D-printed compliant mechanisms, including bi-stable and multi-stable designs, utilizing passive sensing instead of integrated sensors [12,45–47]. These prototypes use stimuli-responsive materials, including liquid crystalline elastomers, conductive polymers and hydrogels and shape-memory polymers (SMPs), [3,6,48–55] to react to environmental

stimuli changes with a higher sensitivity and response time compared to the integrated sensor systems [34,56,57]. The sensitivity of the snap-through energy of these mechanisms could be controlled by specific features of 3D printing [27,34,58]. In addition to these printed tunable parameters, a bistable shape-memory strip was also developed exhibiting thermally sensitive volume changes to the surrounding. Various configurations of these constructs resulted in multi-trigger multi-state mechanisms [59]. A programmable, reversible and rapid buckling-induced 4D-printed active skin was developed, that could be controllably actuated in response to uniaxial tensile loading [60].

Variable stiffness is another recently developed technique for controlling the compliance of 4D-printed systems by heterogeneous SMPs. In other words, the adjustable stiffness of these structures introduces passive sensing that regulates the stiffness of the body according to environmental variations, e.g., humidity and temperature. A sweating actuator was 4D-printed by poly-N-isopropylacrylamide (PNIPAm) and covered with a microporous polyacrylamide (PAAm) dorsal layer. This system operates based on thermoregulation, that is, below the glass transition ($T_g$) temperature the pores are sealed and the actuator is rigid, while above $T_g$, the dilation of the pores causes the sudden decrease in stiffness to soften the stiffness of the fluidic elastomeric actuator [61] (Figure 2c). Bioinspire microneedle arrays with backward barbs were 4D-printed to provide the more sustained use of minimally invasive microneedles for drug delivery and biosensing applications [62]. The 4D-printed system is privileged for the light sensitive barbs that adjust their stiffness based on the environmental changes, for pain-free and ease-of-use functions.

The autonomous adjustment of materials' stiffness could be implemented through various techniques, including thermal stiffness control, acoustic-based control, jamming-based control either in pneumatic or fluidic medium, and electrically or magnetically viscosity-based control [63]. These techniques have shown a great compliance control in enabling the manipulation and grasping of fragile objects [64]. A finger-like soft actuator was 4D-printed using a dielectric elastomer (DE) with an FDM-based 3D printer. The system was modeled via the finite element model (FEM) and used as a soft robotics gripper [65].

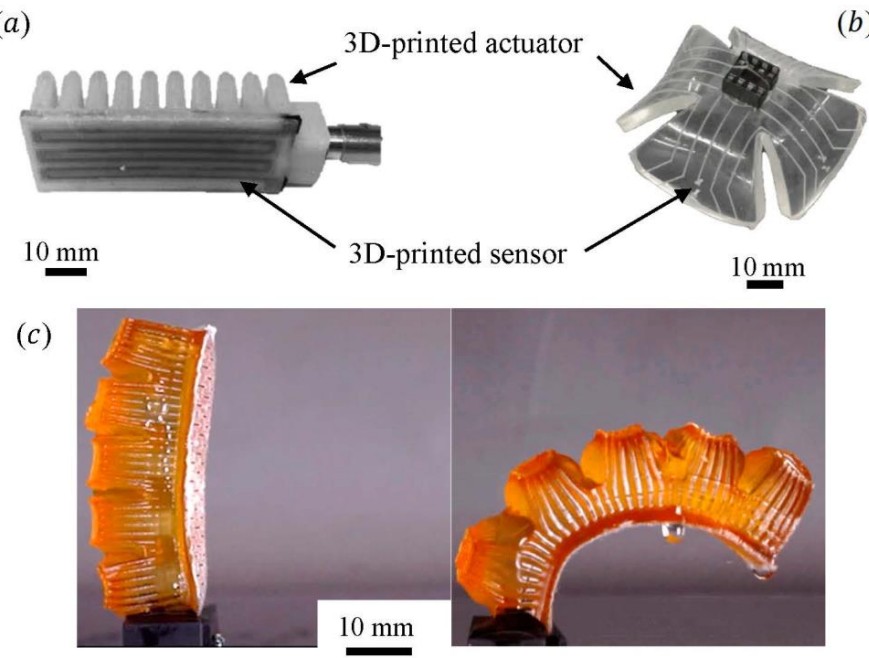

**Figure 2.** *Cont.*

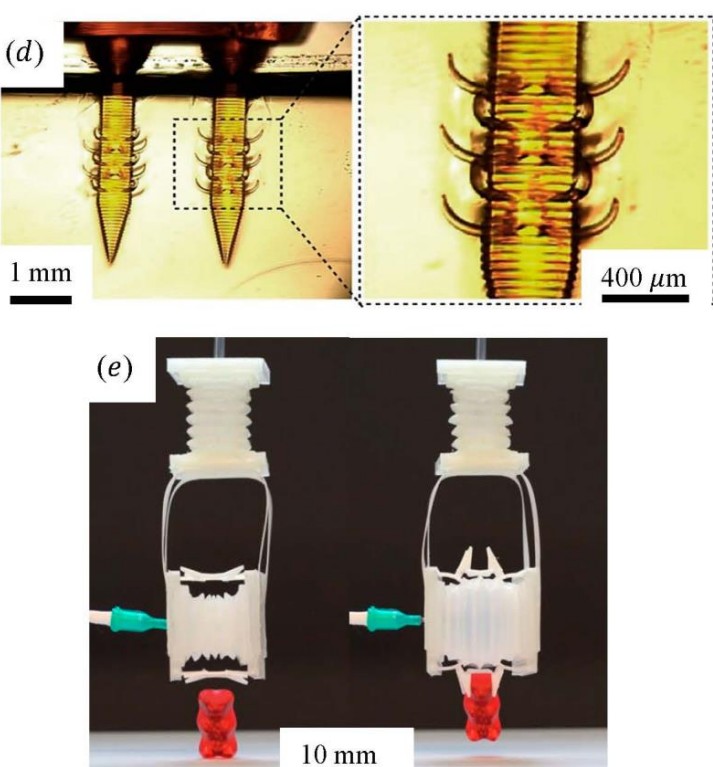

**Figure 2.** Controllable 4D-printed systems (**a**) A combined sensor and 3D-printed pneumatic muscle (reproduced from [66] with the permission of Frontier under Creative Commons CC-BY licence); (**b**) An embedded piezoelectric sensor in the hydrogel actuator (reproduced [67] with the permission of SPIE); (**c**) A sweating 4D-printed system with thermoregulation pores (reproduced from [61] with the permission of The American Association for the Advancement of Science); (**d**) The 4D-printed bioinspired microneedles with stiffness control (reproduced from [62] with the permission of John Wiley and Sons); (**e**) A buckling-induced 4D-printed active skin (reproduced from [60] with the permission of the Nature Publishing Group).

## 3. Integration of 3D-printed Sensors into 4D Printing

The key to developing adaptive 4D-printed systems is to add 3D-printed sensors into 4D printing, which require feedback information, including mechanical motions and deformations and other environmental measurements to realize their environmental interactions. Thus, the 4D-printed system should be equipped for various physical and mechanical properties measurements such as stress, strain, deformation and acceleration to send the required proprioceptive feedback information to the controller unit upon environmental changes. However, the main challenge is to preserve the balance between the conformity of the system and the accuracy of the sensros. The recent progress in the development of functional materials and 3D-printing technology offered accessability to a broad range of sensors with high flexibility and customisation to be integrated into 4D-printed systems for specific uses. In this section, the different types of sensors and sensing principles, which could be employed for prospective applications in adaptive 4D-printed systems, is discussed.

### 3.1. Mechanical Motion and Deformation Measurements

Various thermoplastic materials, such as thermoplastic polyurethane (TPU) and polylactic acid (PLA), have been utilized in mixed form with conductive fillers and liquid metals, such as carbon black (CB) [68] and eutectic gallium−indium (EGaIn) [34], to develop strain sensors. These strain gauge sensors were 3D-printed based on the piezo-resistive or capacitive principles. The pressure variation resulting from geometrical changes generates signals indicating measures such as mechanical

deformations including compression, extension, bending and twisting [69,70]. The 3D-printing sensor industry has utilized the electrical conductivity enhancement through the introduction of conductive fillers such as graphene, but such sensors may suffer from degradation or stress–strain hysteresis issues related to conductive materials. This led to a new field of investigations in 3D-printed fibre optic sensors with a direct application in 4D-printed systems because of being lightweight, transparent and inexpensive [71]. Three-dimensional-printed displacement sensors, based on the Hall effect and Eddy-current principles, were fabricated to remotely identify mechanical motions [72]. The higher switching rate, more design flexibility and the wider range of material access are prominent advantages of this type of sensors against its capacitive and resistive counterparts. An inductance sensor was also 3D-printed incorporating ferrofluidic and magnetic materials into the 4D-printed system at room temperature [73,74]. The prototype demonstrated an acceptable success to measure bending as well as axial and lateral strains. The color-based proprioception method was used for the real-time interaction of soft actuators in an uncertain dynamics environment. A color-based sensing approach was developed to instantly translate the dimensional change of the soft structure into changes in color to reflect local deformations [75].

Piezoelectric sensors have also been 3D-printed with a prospect application in 4D-printed systems. Poly(vinylidene fluoride) (PVDF) was studied as a pressure sensor with a piezo-resistive property that could be 3D printed using electric poling-assisted additive manufacturing (EPAM) [76–80]. A 3D-printed graphite/polydimethylsiloxane (PDMS) sensor was also developed with a capability of detecting small forces in the range 3.5–7.5 mN [81]. Fibre Bragg grating (FBG) was attached to a 3D-printed acrylonitrile butadiene styrene substrate to develop a pressure sensor [82]. Another study demonstrated that the variable impedance properties commonly found in ionic hydrogels could be used for pressure-sensing at a broad range of payloads [67].

Tactile sensors were integrated into 4D-printing to mimic artificial skin by sensing the stimuli resulting from external interactions. These stimuli can be stresses and strains resulting from the mechanical interactions or temperature of the object encountered. Recently, tactile sensors were made from FDM 3D-printing of conductive PLA filament. [83,84]. A multiaxial force sensor was FDM-3D-printed as a composite of TPU and carbon nanotube (CNT) with a piezo resistive surface. Despite some hysteresis, the sensor demonstrated sensitivity to small deflections [68]. A composite of Ecoflex and carbon nanotube (CNT) was developed via 3D printing as a flexible capacitive sensor for tactile and electrochemical sensing [85,86]. Stretchable multi-material, functional tactile sensors were 3D-printed onto freeform surfaces [87]. A tactile proximity sensor made of a EGaIn liquid metal alloy was 3D-printed onto a prosthetic hand for electromyography (EMG) purposes [34].

### 3.2. Environmental Measurements

Temperature and humidity are two essential properties to measure in 4D-printed systems interacting with the external environment. Functional polymers, due to their variable mechanical and electrical response to external stimuli such as humidity, temperature, pH and stress, are often used in 4D printing. In an adaptive 4D-printed system, sensing can be done in spatial layers using integrated sensors [88]. Electrically conductive extrudates, which contain conductive nano particles, have frequently been used in capacitive-based sensor 3D printing for measuring humidity and temperature [89].

Chemical sensors operating in polyelectrolyte solutions were 3D-printed, which demonstrated reversible bending through the polarity changes of the electrode [48]. Three-dimensional printing enabled the calibration through the monitoring of electrolyte ion concentrations. There are also some studies on FDM-printed graphene-based PLA electrodes for electrochemical sensing purposes [90,91].

A conductive composite of PVDF and the multi-walled carbon nanotube (MWCNT) was prepared as the filament feed for the 3D printing of a chemoresistive sensor responsive to volatile organic compounds [92]. The 3D-printed sensor could have applications in adaptive 4D-printed systems in rescue and mine missions. A biological tissue-compatible pH sensor was 3D-printed for wet medium soft

robotics applications. The sensor was demonstrated as operational under cyclic torsional and bending stresses and was made from hydrophilic polyurethane and poly(3,4-ethylene dioxythiophene) [93]. Biocompatible hydrogel sensors were also successfully 3D-printed for drug release and medical diagnostic applications [94–97]. The material properties, geometry and dimensions of the print played a significant role in the performance of the sensors in the adaptive 4D-printed systems to produce a detectable response (Table 1). Each approach of embedding sensors has advantages and disadvantages. These are discussed in detail in earlier papers [4,98].

**Table 1.** Sensors integration into 4D printing.

| Proprioceptive Feedback | Sensors Type | Mechanisms | 3D Printers | Materials | Applications |
|---|---|---|---|---|---|
| *Mechanical Motions and Deformations Measurements* | Stress | Capacitive [99] Optical FBG [82] Piezoelectric [80] | Extrusion FDM EPAM | Ionic gel ABS PVDF | Grasping Tracking Holding Manipulation |
| | Strain | Capacitive [100] Optical waveguide [101] Resistive [102] | Extrusion FDM FDM | Silicone OrmoClear TPU | |
| | Displacement | Eddy current [103–105] Hall effect [72] Optical waveguide [106] | FDM FDM Inkjet | ABS/Copper ABS/Magnetite InkEpo/InOrmo | |
| | Tactile | Piezo-resistive [68] Capacitive [107] | FDM FDM | TPU/CNT TPU | |
| *Environmental Measurements* | Bio | Bioluminescent [94] Electrochemical [108] Vibratory [109] | FDM SLA DLP | ABS/PLA PEGDA Bisphenol | Detection Classification Adaptation |
| | Chemical | Chemoresistive [92] Electrochemical [91] Optical waveguide [110] | FDM FDM SLA | PVDF/MWCNT PLA/Graphene Accura®60 | |
| | Humidity | Solvatochromic [88] | Extrusion | Cu(II)–Thymine | |
| | Temperature | Capacitive [111] | DLW | Nanocrystals | |

## 4. Adaptive 4D-Printed Systems Design

The synergic integration of the main components to develop adaptive 4D-printed systems are explained in this section (Figure 3). In addition to the incorporation of 3D-printed sensors and 4D printing, controller units are required to command the necessary input to the 4D-printed system based on the acquired information. The 4D-printed systems developed so far are mainly reliant on the morphology of soft materials rather than the sophisticated control methods. However, considering the potential exposure to a wide range of dynamic environments, 4D-printed systems demand more robust controllers, particularly in highly sensitive tasks.

A closed-loop controller for the 4D-printed soft robots was recently introduced in order to improve the performance of the soft robots [112]. The active control of 4D prints was achieved by combining the 3D-printed shape-memory polymer composites with a controller that regulated power input to the composite in order to manipulate the material's heating behavior, considering the variations in resistance caused by the changes in strain or temperature [113]. However, the dynamic modeling of such systems was not simply due to their non-linear behavior, infinite degrees of freedom, hyperelasticity, heterogeneous materials properties and hysteresis [114]. In other words, predicting the motion of the actuators based on the input-stimuli could not readily be calculated via the inverse kinematics equations used in the rigid bodies such as the Cosserat rod theory [115] beam theory [116] and constant curvature model [117] methods, because of the computational expenses and steady state assumptions [118]. Therefore, machine-learning methods could be an appropriate choice to realise adaptive 4D-printed systems [119].

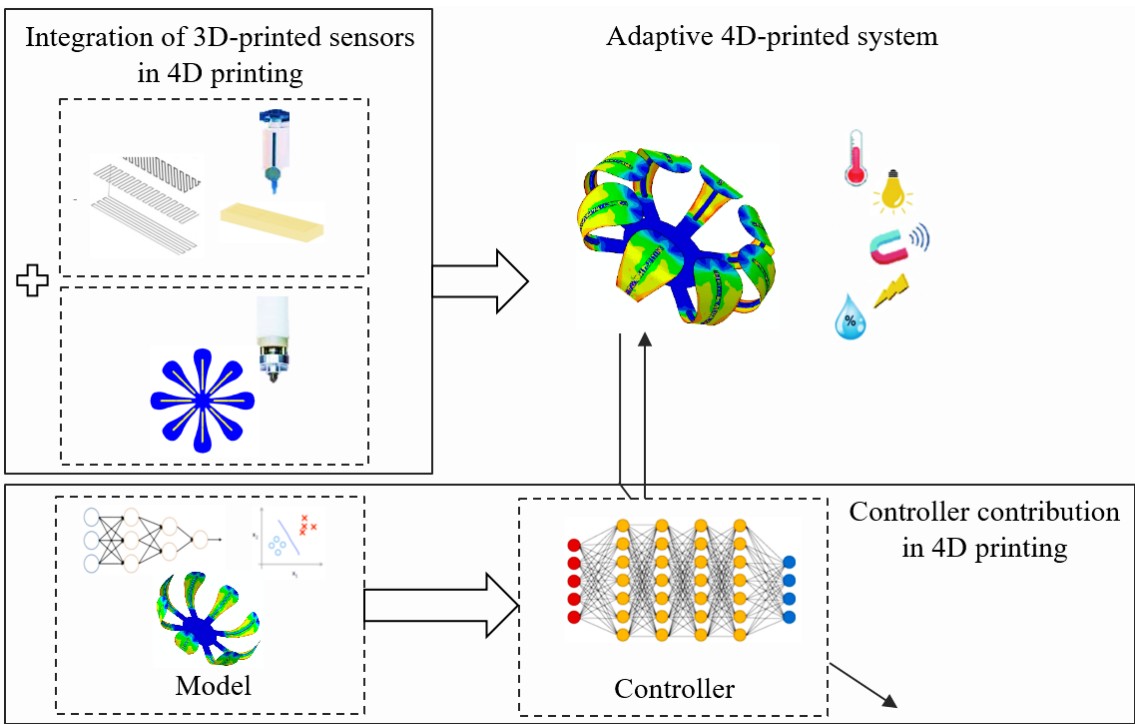

**Figure 3.** Adaptive 4D-printed systems composition.

To equip the 4D-printed systems with adaptive controllability required for real-world dynamic variations, ML algorithms are suitable solutions due to recent advancements in nonlinear systems modeling [120–122]. The finite element model (FEM) is employed to train the ML algorithm to effectively calculate volumetric properties such as spatially heterogeneous mechanical strengths, variable stiffness and the targeted anisotropy during the 4D-printing process. Further, the control algorithms will be developed and incorporated into the 4D-printing platform to compute the precise actuation signal to adapt to the uncertain and dynamic environments [123–126]. However, closed-loop controllers are preferred here to make the most of the integrated 3D-printed sensors in handling uncertainties for wide frequency bandwidths [122,127–130]. Indeed, in order to increase the efficiency of control strategies in terms of time and computational costs, model-based ML [131] in conjunction with self-learning controllers [132,133] are preferred compared to the model-free controllers to deal with the complexity involved in diverse scenarios [66,129,134–136]. A forward dynamics model using recurrent or convolutional neural networks could be employed to implement a model-based feedback controller for an adaptive 4D-printed system [137]. Then, a policy-based reinforcement-learning feedback controller can be used to learn the nonlinear model of a 4D-printed soft robot through experiments and simulation data to compensate for the uncertainties. The self-learning algorithms play a significant role in adaptive 4D-printed systems to optimise the controller commands based on the information acquired from the interaction with the environment via the 3D-printed sensors [138–142].

Model-based ML algorithms, however, require a large pool of data containing different scenarios of environmental changes to be trained to represent the 4D-printed system model with time-variable properties and multi-dimensional control states [143,144]. The training process could be repetitious and expensive if it merely relies on experimental tests, therefore, the FEM (Figure 4) can be used in the offline loop for constructing a comprehensive and reliable model exploring the various possible scenarios [143–149].

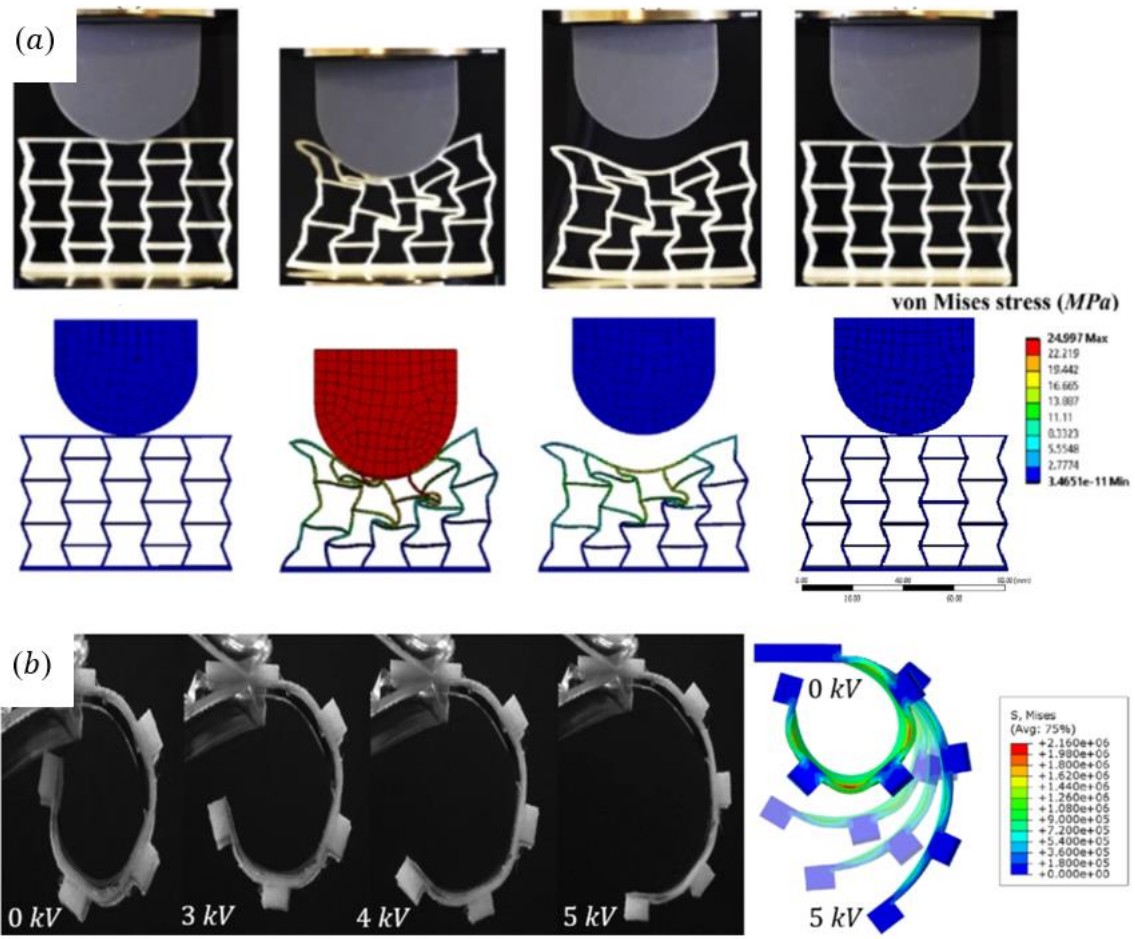

**Figure 4.** Finite element model (FEM) of 4D-printed systems (**a**) A reversible energy absorbing metastructure (reproduced from [27] with the permission of Elsevier); (**b**) A dielectric elastomer (DE) soft gripper (reproduced from [65] with the permission of Elsevier).

Recent advancements in control-based 4D printing with integrating the sensors, ML models trained by FEM results and controlling the entire 4D-printed systems are gathered in Table 2. To the best of the knowledge of the authors, there has yet to be an entirely adaptive 4D-printed system, as defined in this paper. However, there are some promising studies in integrated 3D-printed sensors in 4D printing [7,16,38–40,67,150], the application of the FEM [151] and ML-based controllers in 4D-printed systems [34,66,135]. These relevant research results support the possibility of realizing the proposed adaptive 4D-printed systems.

**Table 2.** Studies toward adaptive 4D-printed systems.

| Refs. | 4D Printing Equipped with Sensors | Controller | FEM |
|:---:|:---:|:---:|:---:|
| [34] | ✓ | ✓ | |
| [66] | ✓ | ✓ | |
| [75] | ✓ | ✓ | |
| [135] | ✓ | ✓ | |
| [56] | ✓ | | ✓ |
| [57] | ✓ | | ✓ |
| [61] | ✓ | | ✓ |
| [63] | ✓ | | ✓ |
| [65] | ✓ | | ✓ |
| [152] | ✓ | | ✓ |
| [153] | ✓ | | ✓ |
| [7] | ✓ | | |
| [16] | ✓ | | |
| [38] | ✓ | | |
| [39] | ✓ | | |
| [40] | ✓ | | |
| [67] | ✓ | | |
| [150] | ✓ | | |

Additionally, there have been several studies on the design and fabrication of 4D-printed compliant mechanisms promoting controlled self-management and self-actuation without needing sensors [34,56,63,152–154]. Further investigations on equipping such mechanisms with controllers to provide more flexibility and adaptability are envisaged, allowing for their wider ranges of autonomous operations. It is interesting to note that, there have been only a few studies on the use of the FEM to design and optimize controllers in 4D-printed systems. This will certainly increase the robustness and repeatability of 4D-printed systems in diverse applications with variable unstructured environments.

## 5. Discussions and Future Perspectives

The three-dimensional printing of sensors and actuators have provided a new platform for developing mass-customized systems with autonomous capabilities owing to recent advancements of ML-based control algorithms. This paper has proposed and discussed the new technology of adaptive 4D-printed systems which are printed systems that incorporate 3D-printed sensors and control algorithms and programmed to handle delicate tasks in dynamic environments.

It was emphasized that the synthesis and processing of stimuli-responsive materials played a key role for the development of entirely adaptive 4D-printed systems. The recent advancements of 3D-printed sensors for measuring feedback including mechanical motions and deformations as well as physical and chemical proprioceptive information and their specific applications to adaptive 4D-printed systems were discussed. ML-based control algorithms were suggested and reviewed to compensate for the uncertainties pertaining to 4D-printed systems, which originated from non-linear mechanical characteristics and the viscoelastic nature of soft polymeric materials. Although there are materials available to achieve adaptive 4D-printed systems, there is a scope for further investigations on 3D-printable mechanochromic materials that exhibit variable mechano-responsive properties in different environmental conditions. Moreover, promoting the further applications of adaptive 4D-printed systems in key industries, such as automotive, agriculture and aerospace, demands stimuli-responsive and printable materials with a high thermal and chemical stability to function in corrosive, high-temperature and high-pressure, harsh environmental conditions. Piezoresistive and piezoelectric materials, such as PVDF, have exhibited suitable characteristics for integrated sensors and actuators. However, further investigations are required to enhance their printability and piezoelectric performances.

Four-dimensional printing provides a situation for using a random and/or free source of energy in nature to make non-random structures [155]. Studies have shown that energy consumption in 4D printing is the least of all manufacturing processes [14]. Four-dimensional printing could utilize abundant biodegradable materials in nature, for example, chitin from fisheries and cellulose from plants as the most abundant and broadly distributed natural materials in the ecosystem [48]. The current 4D-printed systems do not possess adaptability and longevity and hence are not environmentally friendly. However, the proposed adaptive 4D-printed systems aim to produce systems with a good stability and repeated use, benefiting from the recyclability and low carbon footprint of bio and organic stimuli-responsive polymers.

There is still more demand for developing the multiphysics simulation and control platforms for 4D printing to accelerate and support the practical applications of adaptive 4D-printed systems in diverse environmental interactions [156–158]. The reversible multi-stable compliant mechanisms should be incorporated into 4D printing as sensorless adaptive 4D-printed systems [159]. Four-dimensional printing is often accompanied by considerable bending that may lead to the rupture of particularly soft materials. Therefore, the development of self-healing and shape-changing materials can be one of the future directions of development. The amphibian 4D-printed systems with the adaptability to operate in various ambiances is another direction of future research [160].

They are still unsolved problems in practical applications of 4D-printed systems, particularly in the high-frequency bandwidth in terms of control complexities. Hence, embedding the controller in the volumetric pixel (voxel) performing as the controller unit with zero-lag feedback based on the morphological properties could be promising approach for faster and intuitive control. Another approach could be the optimal placement of sensors and actuators via 3D printing as voxel arrays to significantly impact controller performance.

The 4D-printed systems scalability is another focus of the research due to the challenges associated with making microscale features, like channels and voids [161]. Miniaturizing 4D-printed systems to include millimetre-sized features with an adaptability to dynamic environments open new applications of these products in microfluidics and cells manipulations. This requires further study on the materials aspect of 4D printing, for enhancing the controlled morphological performance of stimuli-responsive polymers as both sensor and actuator units [103,162,163], whilst considering the current resolution limits and technology of 3D printers.

## 6. Conclusions

Current additive manufacturing is not about producing a fully controllable dynamic product, but mainly about intricately designed static structures. Four-dimensional (4D) printing was recently introduced as dynamic interactive systems with limited applications. This paper introduced the control-based 4D printing that is adaptive 4D-printed systems with the synergic incorporation of 3D-printed sensors and machine learning (ML)-based controllers into 4D printing, leading to the harnessing of more useful applications for 4D printing in dynamic and uncertain environments.

Different types of 4D-printed systems were presented and reviewed. The integration of three-dimensional (3D)-printed sensors into 4D printing and its applications and benefits were discussed. The need for the development of adaptive 4D-printed systems and the techniques for the modeling and control of them were explained. The role of the finite element model (FEM) in developing models and designing the ML-based controllers were described. The studies towards control-based 4D printing for realizing adaptive 4D-printed systems and their current challenges were outlined. The necessity of adaptive 4D-printed systems and their versatile applications require further improvements of 3D-printable stimuli-responsive materials in terms of sensitivity, strength and stability, as well as more a precise FEM, and faster voxelized controllers based on the morphological properties. Developing such systems will open possibilities for innovative applications in medical, manufacturing and agricultural fields.

**Author Contributions:** Conceptualization, A.Z.; methodology, A.Z., M.B..; investigation, A.Z., M.B., S.G., A.K. and A.Z.K.; resources, S.N.; writing—original draft preparation, A.Z., A.K., M.B. and S.G.; writing—review and editing, A.K., A.Z.K., M.B., S.G. and A.Z.; supervision, A.K and S.N.; project administration, A.Z. All authors have read and agreed to the published version of the manuscript.

**Funding:** This research received no external funding.

**Conflicts of Interest:** The authors declare no conflict of interest.

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
