# Peer review of "Control-Based 4D Printing: Adaptive 4D-Printed Systems"

_applsci, doi:10.3390/app10093020_

Round 1
Reviewer 1 Report
Well organized and written work. Presents a review of the literature on an interesting and modern issue.
I noted some remarks in the attached file.

Author Response
Reviewer: 1
Well organized and written work. Presents a review of the literature on an interesting and modern issue.
I noted some remarks in the attached file.
Response: Thank you for acknowledging our work and suggestions provided in the peer-review file. The suggestions were addressed in the revised manuscript.

Reviewer 2 Report
In this manuscript, the recent progress of adaptive 4D printing has been reviewed. The authors introduced controllable 4D-printed systems, integrated sensing systems, and adaptive system designs. The challenges and limitations of the technology were also discussed. The paper could be accepted for publication after minor revision.
The main concern of this review paper is the connections between each section. I strongly encourage authors to add paragraphs and sentences to discuss and clarify how the paper has been structured. These additions could also help readers to more clearly understand the flow the paper.
Additionally, please double check the details of figures and text, for example, the address on Line 5 and the dashed frame in Figure 1.
Author Response
Reviewer: 2
In this manuscript, the recent progress of adaptive 4D printing has been reviewed. The authors introduced controllable 4D-printed systems, integrated sensing systems, and adaptive system designs. The challenges and limitations of the technology were also discussed. The paper could be accepted for publication after minor revision.
The main concern of this review paper is the connections between each section. I strongly encourage authors to add paragraphs and sentences to discuss and clarify how the paper has been structured. These additions could also help readers to more clearly understand the flow of the paper.
Response: Thank you for your suggestion. A new paragraph is added to the revised version of the manuscript providing more cohesion storyline of the manuscript as follows:
“This paper consists of four sections as follows: introduction of adaptive 4D-printed systems and various methods to incorporate control strategies into 4D printing is described in Section 2. It is followed by analyzing the integration of 3D-printed sensors into 4D printing in Section 3. This section explores different types of sensors that could be used in control-based 4D printing to provide the proprioceptive and environmental feedback information. In Section 4, the adaptive 4D-printed systems design are explained. Section 5, concludes the paper by presenting the challenges and future opportunities of these systems.”
Additionally, please double-check the details of figures and text, for example, the address on Line 5 and the dashed frame in Figure 1.
Response: The mentioned points are corrected in the revised version of manuscripts.
